systems biology/computational biology/bioinformatics

cancer cell metabolism, flux balance analysis, head and neck squamous cell carcinoma, multiple-level optimization

**Author for correspondence:**
Feng-Sheng Wang
e-mail: chmfsw@ccu.edu.tw

# Oncogene inference optimization using constraint-based modelling incorporated with protein expression in normal and tumour tissues

Wu-Hsiung Wu[1], Fan-Yu Li[1], Yi-Chen Shu[1], Jin-Mei Lai[2], Peter Mu-Hsin Chang[3,4], Chi-Ying F. Huang[5,6] and Feng-Sheng Wang[1]

[1]Department of Chemical Engineering, National Chung Cheng University, Chiayi, Taiwan
[2]Department of Life Science, Fu Jen Catholic University, New Taipei City, Taiwan
[3]Department of Oncology, Taipei Veterans General Hospital, Taipei, Taiwan
[4]Faculty of Medicine, [5]Institute of Biopharmaceutical Sciences, and [6]Department of Biotechnology and Laboratory Science in Medicine, National Yang-Ming University, Taipei, Taiwan

F-SW, 0000-0001-5266-2346

Cancer cells are known to exhibit unusual metabolic activity, and yet few metabolic cancer driver genes are known. Genetic alterations and epigenetic modifications of cancer cells result in the abnormal regulation of cellular metabolic pathways that are different when compared with normal cells. Such a metabolic reprogramming can be simulated using constraint-based modelling approaches towards predicting oncogenes. We introduced the tri-level optimization problem to use the metabolic reprogramming towards inferring oncogenes. The algorithm incorporated Recon 2.2 network with the Human Protein Atlas to reconstruct genome-scale metabolic network models of the tissue-specific cells at normal and cancer states, respectively. Such reconstructed models were applied to build the templates of the metabolic reprogramming between normal and cancer cell metabolism. The inference optimization problem was formulated to use the templates as a measure towards predicting oncogenes. The nested hybrid differential evolution algorithm was applied to solve the problem to overcome solving difficulty for transferring the inner optimization problem into the single one. Head and neck squamous cells were applied as a case study to evaluate the

algorithm. We detected 13 of the top-ranked one-hit dysregulations and 17 of the top-ranked two-hit oncogenes with high similarity ratios to the templates. According to the literature survey, most inferred oncogenes are consistent with the observation in various tissues. Furthermore, the inferred oncogenes were highly connected with the TP53/AKT/IGF/MTOR signalling pathway through PTEN, which is one of the most frequently detected tumour suppressor genes in human cancer.

## 1. Introduction

Metabolic fluxes are altered in cancer cells and differ from the fluxes in the healthy situation. Normal cells primarily produce energy through mitochondrial oxidative phosphorylation. However, cancer cells undergo abnormal metabolism and use glucose differently than normal cells [1–4]. Cancer cells become habituated to certain fuel sources and metabolic pathways (metabolic reprogramming), profoundly changing how they consume and use nutrients such as glucose. Such flux reprogramming in cancer metabolism is characterized by high glucose consumption and lactate production, even under aerobic conditions, as well as increased glutamine catabolism and amino acid metabolism [1,2]. This is also termed as the Warburg effect [1–4]. Many recent studies on cancer metabolism have focused on the regulation of metabolic enzyme expression by oncogenes and tumour suppressors to identify multiple cancer-associated genes and pathways.

Human metabolism is complex and specialized in different tissue and cell types. Mapping out these tissue-specific metabolisms in genome-scale models will provide deeper insights into the metabolic basis of various physiological and pathological processes. Constraint-based model (CBM) approaches have been applied to predict oncogenes, essential enzymes and drug targets for developing novel medical treatments [5–11]. Flux-dependent and pruning methods are applied to reconstruct tissue-specific genome-scale metabolic models for predicting tissue-specific behaviours [6,10]. Schultz & Qutub [12] recently introduced cost optimization reaction dependency assessment (CORDA) to develop concise, but not minimalistic, tissue-specific metabolic models based on omics data and a generalized human metabolic reconstruction. The development of genome-scale human metabolic networks, such as Recon 1 [13], Recon 2 [14] and human metabolic reactions [8,15], has resulted in the emergence of network medicine. The recently updated and extended Recon 2 (i.e. Recon 2.2) and Recon 3D are the most comprehensive human genome-scale network reconstruction models [16,17]. Recon 3D [16] includes three-dimensional metabolite and protein structure data and enables the integrated analyses of metabolic functions in humans. Recon 2.2 is used to balance all reactions and improve the simulation of energy metabolism [17]. Recently, Richelle *et al.* [18] built 44 different genome-scale metabolic models from Recon 2.2 [17] and iHsa [19] using six model extraction methods with RNA-Seq data from NCI-60 cell line. Such an approach provides guidelines for the development of the next-generation of data contextualization methods. Ryu *et al.* [20] presented a systematic framework for the generation of gene-transcript-protein-reaction associations in the human metabolism and addition of new reactions from Recon 2.2 to build Recon 2M.2 that is biochemically consistent and transcript-level data compatible. Such gene-transcript-protein-reaction information enabled more accurate simulation of cancer metabolism and prediction of anticancer targets.

Wu *et al.* [21] developed a CBM based on the published Recon 2 and the Warburg effects of mouse hepatocytes deficient in miR-122a, and inferred that *DDC* is an oncogene. A total of 38 metabolic profiles obtained using LC/MS from the liver tissues of 10 control mice and 10 Mir122a$^{-/-}$ mice were applied to CBM for evaluating similarity ratios between the deficient and normal states. However, genome-wide information is not used in this approach, and the changes in metabolite concentrations may not be equal to the flux-sum alteration. In the present study, we established an algorithm to incorporate the Recon 2.2 network [17] with the Human Protein Atlas (HPA) database [22] and used the CORDA to reconstruct genome-scale metabolic network (GSMN) models of head and neck squamous cells (HNSCs) at healthy and cancer states. The models were then applied to build templates of flux alterations between cancer and normal cells.

A tri-level inference optimization framework integrated the templates and CBM was developed to infer dysregulated enzymes that contribute to inducing head and neck squamous cell carcinoma (HNSCC). Such framework can also be used to mimic gene screening procedures in wet laboratory for detecting dysregulated oncogenes. A tri-level optimization problem (TLOP) integrating splice-isoform expression has been introduced to depict breast cancer metabolism [23]. This study

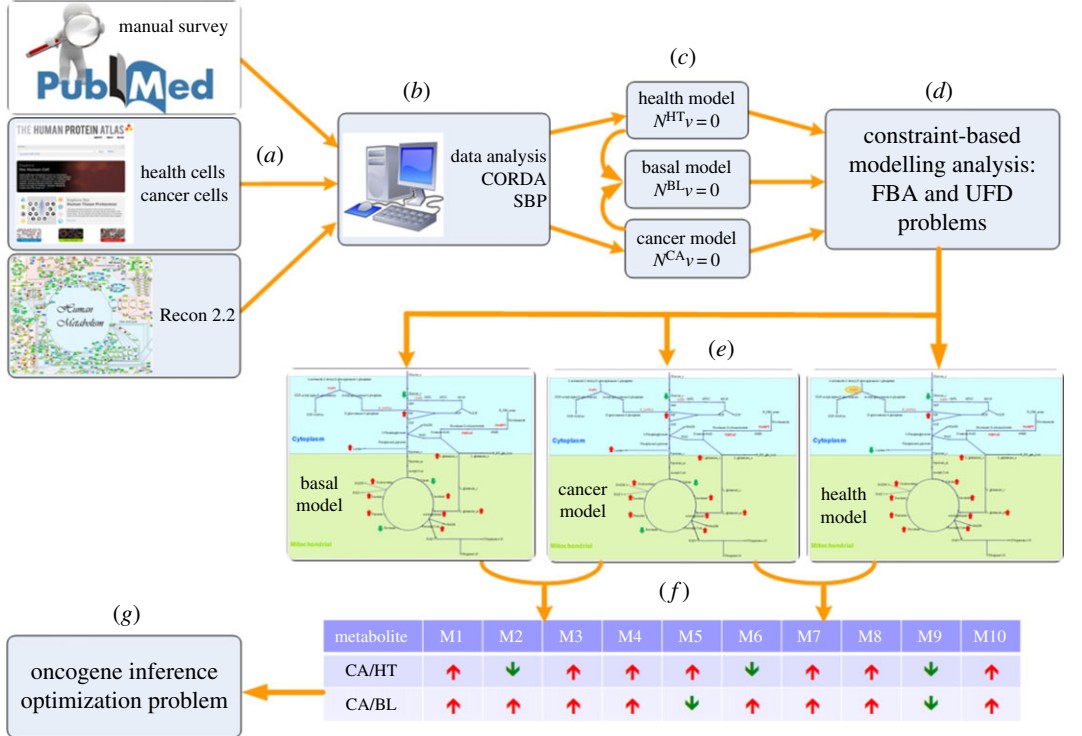

templates of flux-sum alterations for CA/HT and CA/BL models

**Figure 1.** Templates of flux-sum alterations. (*a*) Procedure for survey and collection of information for head and neck squamous cells from PubMed, HPA and Recon 2.2. (*b*) Develop cancer (CA), healthy (HT) and basal (BL) specific models using CORDA and SBP algorithms. (*c*) The basal (BL) model, a union set of both models in (*b*), was used to investigate how healthy cells can smoothly detour the metabolic reprogramming to that of cancer cells. (*d*) Constraint-based methods were applied to the flux distributions for each model. (*e*) Flux-sum distributions were computed for each model. (*f*) Two templates of flux-sum alterations for CA related to HT and CA to BL models were generated. (*g*) Detect oncogenes by solving the oncogene inference optimization problem using the template similarity as the objective. The up-arrow/down-arrow notation indicates an increasing/decreasing flux-sum of the *i*th metabolite in cancer state.

introduced a similarity measure in the TLOP to decide mutated genes and their corresponding dysregulated bounds. The similarity between mutant flux patterns and templates of flux alterations was used as the objective. Duality theory is generally incapable of transforming inner problems into single-level problems. Solving the tri-level problem is difficult. We introduced a nested hybrid differential evolution (NHDE) algorithm for solving the TLOP to detect dysregulated oncogenes. The similarity measure was provided for NHDE to evolve new mutants for achieving higher ranked oncogenes.

# 2. Methods

## 2.1. Templates of flux-sum alterations for normal and cancer cells

This study introduced six steps (figure 1) to establish GSMN models for cancer (CA), healthy (HT) and basal (BL) models of head and neck squamous cells (HNSC) and to build their corresponding templates of flux-sum alterations. The BL model refers to the normal model, which represents normal situations of head and neck squamous tissues. PubMed literature survey, the HPA database [22] and the Recon 2.2 human metabolic network [17] were used to generate the specific information for HNSC (figure 1*a*,*b*). We first retrieve gene expression using the gene association of Recon 2.2 and the coding protein expression in normal and tumour tissues from the HPA database. Applying dependency assessment, the reactions having such expressions are identified into high, medium and negative confidence groups. The whole metabolic network of Recon 2.2 and the dependence reactions are provided as the input information for the CORDA algorithm [12] to reconstruct a

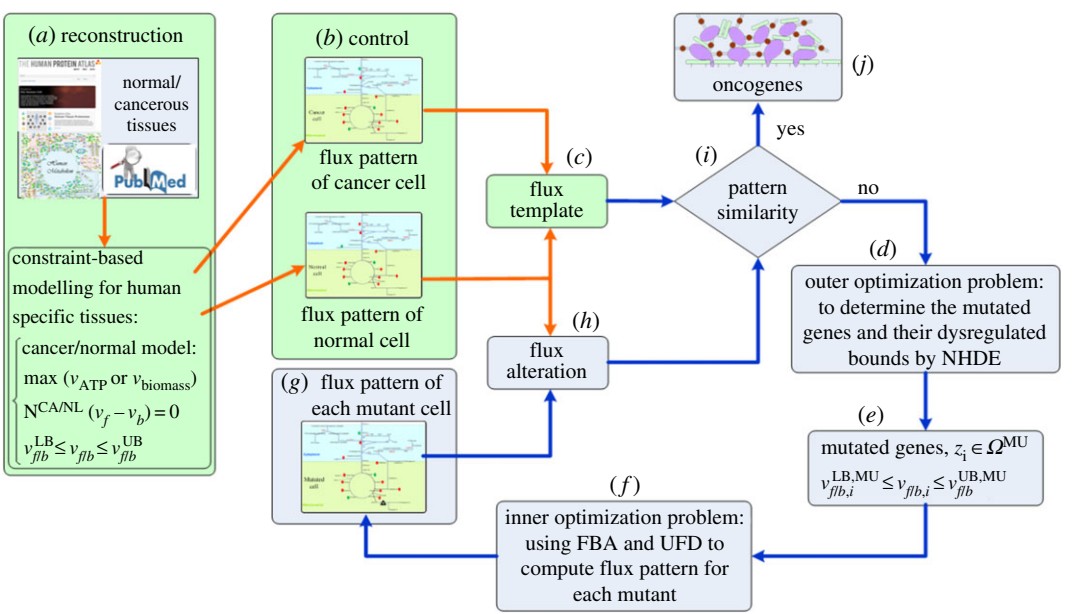

**Figure 2.** Oncogene inference problem incorporated with the templates. (*a*) Reconstruct the cancer and normal metabolic networks of colon cells using databases, such as HPA and Recon 2.2, and build the constraint-based model for the networks. (*b*) Compute the flux distributions of cancer and normal models. (*c*) Build the flux template according to the flux distributions of cancer and normal models for the control of oncogene inference. (*d–j*) Blue arrows present a tri-level optimization problem to mimic mutant schemes in a wet laboratory for oncogene inference. (*d*) The outer optimization problem is employed to decide modulated genes by NHDE. (*e*) A set of mutated genes and their corresponding dysregulated bounds are obtained. (*f*) The inner optimization problem consisting of flux balance analysis (FBA) and uniform flux distribution (UFD) problems computes the flux distribution for each mutant. (*g*) Each mutant flux pattern is obtained from FBA and UFD. (*h*) Use the mutant flux patterns to evaluate flux alteration of each mutant. (*i*) Compare flux template and flux alteration until the pattern similarity satisfied; otherwise, repeat the computation. (*j*) Obtain the optimal oncogenes.

tissue-specific metabolic network (figure 1*b*), which consists of a thousand reactions and species. We have developed a systems biology program (SBP) tool to automatically build cancer and healthy stoichiometric models (figure 1*c*) in the GAMS programming language. The basal model (union set of CA and HT models) was used to investigate how the healthy cell could smoothly detour metabolic reprogramming to cancer. CBM approaches were then applied to analyse the structure of the models and to compute flux distributions for each model (figure 1*c,d*). Such flux distributions were applied to build the templates of flux-sum alterations for CA to HT models and CA to BL models (figure 1*f*). For example, flux-sum alteration of metabolite M1 increased between CA and HT model, but decreased for metabolite M2 as shown in figure 1*f* .

## 2.2. Tri-level optimization for inferring oncogenes

The oncogene inference framework (figure 2) integrated the templates and the TLOP was used to simulate a wet laboratory to detect dysregulated oncogenes in HNSC. The first, second and third procedures in figure 2 are discussed in figure 1 to build the flux template, which acts as a control in the oncogene inference problem. GSMN models of cancerous and normal cells were reconstructed as shown in figure 2*a*. Both models were then applied to compute the flux distribution patterns at the cancer and normal situations (figure 2*b*). The flux template was built according to the flux distributions of cancer and normal models (figure 2*c*). Blue arrows (figure 2*d–j*) present the TLOP to mimic mutant schemes in a wet laboratory for oncogene inference. The outer optimization problem was employed to decide mutation genes and their corresponding dysregulated bounds. The decision variables were used in the flux balance analysis (FBA) and uniform flux distribution (UFD) problems to hierarchically compute flux distribution for each mutant. The fluxes obtained from the decision variables in FBA and UFD problems were then applied to evaluate the similarity ratios in the outer problem. The evolutionary computation was repeated until

the optimal result was achieved. The TLOP was defined as follows:

$$
\begin{cases}
\text{Outer optimization problem:} \\[4pt]
\max_{\delta, z_i} (\mathrm{SR}^{T,\mathrm{HT}} + \mathrm{SR}^{W,\mathrm{HT}}) + \mathrm{SR}^{T,\mathrm{HT}}\mathrm{SR}^{W,\mathrm{HT}} \\[4pt]
\max_{\delta, z_i} (\mathrm{SR}^{T,\mathrm{BL}} + \mathrm{SR}^{W,\mathrm{BL}}) + \mathrm{SR}^{T,\mathrm{BL}}\mathrm{SR}^{W,\mathrm{BL}} \\[4pt]
\text{subject to the inner optimization problems,} \\
\text{FBA problem:} \\
\quad
\begin{cases}
\max\limits_{\mathbf{v}_{f/b}} v_{\mathrm{biomass}} \\
\text{subject to} \\
\quad \mathbf{N}(\mathbf{v}_f - \mathbf{v}_b) = \mathbf{0} \\
\quad v_{f/b,i}^{\mathrm{LB,\,MU}} \le v_{f/b,i} \le v_{f/b,i}^{\mathrm{UB,\,MU}},\, z_i \in \Omega^{\mathrm{MU}} \\
\quad v_{f/b,j}^{\mathrm{LB}} \le v_{f/b,j} \le v_{f/b,j}^{\mathrm{UB}},\, z_j \notin \Omega^{\mathrm{MU}}
\end{cases} \\
\text{UFD problem:} \\
\quad
\begin{cases}
\min\limits_{\mathbf{v}_{f/b}} \sum\limits_{k \in \Omega^{\mathrm{Int}}} (v_{f,k})^2 + (v_{b,k})^2 \\
\text{subject to} \\
\quad \mathbf{N}(\mathbf{v}_f - \mathbf{v}_b) = \mathbf{0} \\
\quad v_{f/b,i}^{\mathrm{LB,\,MU}} \le v_{f/b,i} \le v_{f/b,i}^{\mathrm{UB,\,MU}},\, z_i \in \Omega^{\mathrm{MU}} \\
\quad v_{f/b,j}^{\mathrm{LB}} \le v_{f/b,j} \le v_{f/b,j}^{\mathrm{UB}},\, z_j \notin \Omega^{\mathrm{MU}} \\
\quad v_{\mathrm{biomass}} \ge v_{\mathrm{biomass,}}^{*}
\end{cases}
\end{cases}
\tag{2.1}
$$

where $\mathbf{v}_f$ and $\mathbf{v}_b$ are the forward and backward fluxes of reversible reactions, respectively; $\mathbf{N}$ is an $m \times n$ stoichiometric matrix, where $m$ is the number of metabolites and $n$ is the number of reactions; $v_{f/b,j}^{\mathrm{LB}}$ and $v_{f/b,j}^{\mathrm{UB}}$ are the positive lower and upper bounds of the $j$th forward/backward flux, respectively; $v_{f/b,i}^{\mathrm{LB,\,MU}}$ and $v_{f/b,i}^{\mathrm{UB,\,MU}}$ are the lower and upper bounds of the $i$th upregulation, downregulation, or knockout flux due to the $i$th enzyme dysregulation, which is determined using the outer optimization problem; $v_{\mathrm{biomass}}^{*}$ is the objective value obtained from FBA; $\Omega^{\mathrm{Int}}$ is the set of internal reactions; and $\Omega^{\mathrm{MU}}$ is the set of mutated reactions. The objective of the FBA is the maximization of the biomass growth rate for the cancer and mutant cells. However, normal cells may have different objectives depending on growth signals [24]; thus, the maximization of ATP synthesis rate $v_{\mathrm{ATP}}$ was applied in this situation.

The reversible reactions were separated into forward and backward reactions, $v_f$ and $v_b$, for building the constraint-based model of HNSC. This representation could directly compute flux-sum distributions for each metabolite, and evaluate the choke-point metabolites through the stoichiometric matrix. We define three categories of choke-point metabolites: a choke-point metabolite connected with a single-ingoing and multi-outgoing reaction, a choke-point metabolite connected with a multi-ingoing and single-outgoing reaction, or a choke-point metabolite connected with a single-ingoing and single-outgoing reaction (electronic supplementary material, S1).

The value for the dysregulated bounds is computed using the following equations:

Upregulation:

$$
\begin{cases}
(1 - \delta)v_{f,i}^{\mathrm{basal}} + \delta v_{f,i}^{\mathrm{UB}} \le v_{f,i} \le v_{f,i}^{\mathrm{UB}} \\
v_{b,i}^{\mathrm{LB}} \le v_{b,i} \le (1 - \delta)v_{b,i}^{\mathrm{basal}} + \delta v_{b,i}^{\mathrm{LB}},\, z_i \in \Omega^{\mathrm{MU}}
\end{cases}
$$

Downregulation:

$$
\begin{cases}
v_{f,i}^{\mathrm{LB}} \le v_{f,i} \le (1 - \delta)v_{f,i}^{\mathrm{basal}} + \delta v_{f,i}^{\mathrm{LB}} \\
(1 - \delta)v_{b,i}^{\mathrm{basal}} + \delta v_{b,i}^{\mathrm{UB}} \le v_{b,i} \le v_{b,i}^{\mathrm{UB}},\, z_i \in \Omega^{\mathrm{MU}}
\end{cases}
\tag{2.2}
$$

Knockout:

$$
v_{f,i} = v_{b,i} = 0,\, z_i \in \Omega^{\mathrm{MU}},
$$

where $v_{f/b,i}^{\mathrm{basal}}$ is the basal flux in the normal state; $\delta$ determined in the outer optimization problem is the regulation strength parameter with a value within (0, 1]. The integer variable $z_i$ is used to determine the dysregulated enzyme, so that the bounds of the modulated reactions are restricted to be backward or forward upregulation, backward or forward downregulation, or knockout reactions, as shown in equation (2.2).

The objective function in the outer optimization problem is a similarity measure that compares the templates of flux-sum alterations for CA with HT and BL models and with the Warburg hypothesis. Here, $\mathrm{SR}^{T,BL/HT}$ and $\mathrm{SR}^{W,BL/HT}$ are defined as the similarity ratios for the templates of flux-sum alterations compared CA with BL/HT models and the Warburg hypothesis obtained from the literature, respectively, and the similarity ratios (electronic supplementary material, S1) are evaluated as follows:

$$\mathrm{SR}^{T/W} = \frac{\sum_{m=1}^{N_{T/W}} |\mu_m^{T/W}|}{N_{T/W}}, \tag{2.3}$$

where the similarity indicator ($\mu_m^{T/W}$) for the $m$th metabolite is defined as follows:

$$\mu_m^{T/W} = \begin{cases} 1, & \text{if } \mathrm{LFC}_m > \mathrm{tol} \text{ and } \mathrm{LFC}_m^{T/W} > \mathrm{tol} \\ -1, & \text{if } \mathrm{LFC}_m < -\mathrm{tol} \text{ and } \mathrm{LFC}_m^{T/W} < -\mathrm{tol} \\ 0, & \text{otherwise}, \end{cases} \tag{2.4}$$

where the logarithmic fold changes in the $m$th metabolite, $\mathrm{LFC}_m^{T/W}$, for the templates or the Warburg hypothesis are provided in advance. Here the tolerance is defined as $\mathrm{tol} = \log_2(1 + \varepsilon)$, and $\varepsilon$ is a percentage of flux alteration. The logarithmic fold change ($\mathrm{LFC}_m$) between the synthesis rates of the $m$th metabolite in cancer/dysregulated (denoted as deficient) and basal/healthy (denoted as normal) states is computed as follows:

$$\mathrm{LFC}_m = \log_2\left(\frac{r_{m,\text{deficient}}}{r_{m,\text{normal}}}\right), \tag{2.5}$$

where the overall synthesis rate ($r_m$) of the $m$th intracellular compound at deficient and normal states is evaluated as follows:

$$r_m = \sum_{i \in \Omega^c}\left(\sum_{N_{ij}>0,j} N_{ij} v_{f,j} - \sum_{N_{ij}<0,j} N_{ij} v_{b,j}\right), m \in \Omega^m. \tag{2.6}$$

Here $\Omega^c$ is the set of metabolites located in different compartments, and $\Omega^m$ is the set of metabolites. The forward and backward fluxes ($v_{f,j}$ and $v_{b,j}$) are obtained from the UFD problem.

The TLOP in equation (2.1) consists of FBA and UFD problems in the inner optimization problem. The objective of FBA is to maximize the biomass formation rate for the cancer model. However, normal cells may have different objectives depending on growth signals [24]; thus, the maximum ATP synthesis rate is applied in this situation. The optimal flux distribution of the FBA problem is generally not unique; a large set of alternative flux distributions with an identical objective value exists. We minimize the squared sum of all internal fluxes for the UFD to ensure the efficient channelling of all fluxes through all pathways to eliminate the multiplicity of flux distributions in equation (2.1). The minimizing Euclidean norm problem is referred to as the UFD problem, which is a quadratic programming problem that has a unique solution.

## 2.3. Nested hybrid differential evolution

It is difficult to convert the inner optimization problems (FBA and UFD) of the TLOP into constraints by using duality transformation. In this study, we extended the NHDE algorithm to solve the TLOP (electronic supplementary material, S1). The computational concept of NHDE (figure 3) is based on hybrid differential evolution (HDE), which was extended from the original differential evolution (DE) algorithm [25–27]. The basic operations of the NHDE algorithm, except for coding representation, selection and evaluation operations, are similar to those of the DE and HDE algorithms. The NHDE algorithm has been applied to solve metabolic engineering [28] and biomedical problems [29,30]. The NHDE algorithm is used to identify integer variables in the outer optimization problem to determine which genes are selected to be regulated, and the inner optimization problems (FBA and UFD) are then solved using a linear and quadratic optimization solver. An optimal solution for each candidate individual is achieved when the FBA problem is convergent, and the set of these individual solutions comprises a feasible solution to the TLOP.

In this study, the NHDE algorithm was implemented in the General Algebraic Modelling System (GAMS) environment, and its performance and solution quality depended on three setting factors: the tolerance ratio used in migration, population size and maximum number of iterations. The crossover factor and tolerance ratio were set to 0.5 and 0.05, respectively. A population size of 50 was used, and the maximum number of iterations was 100.

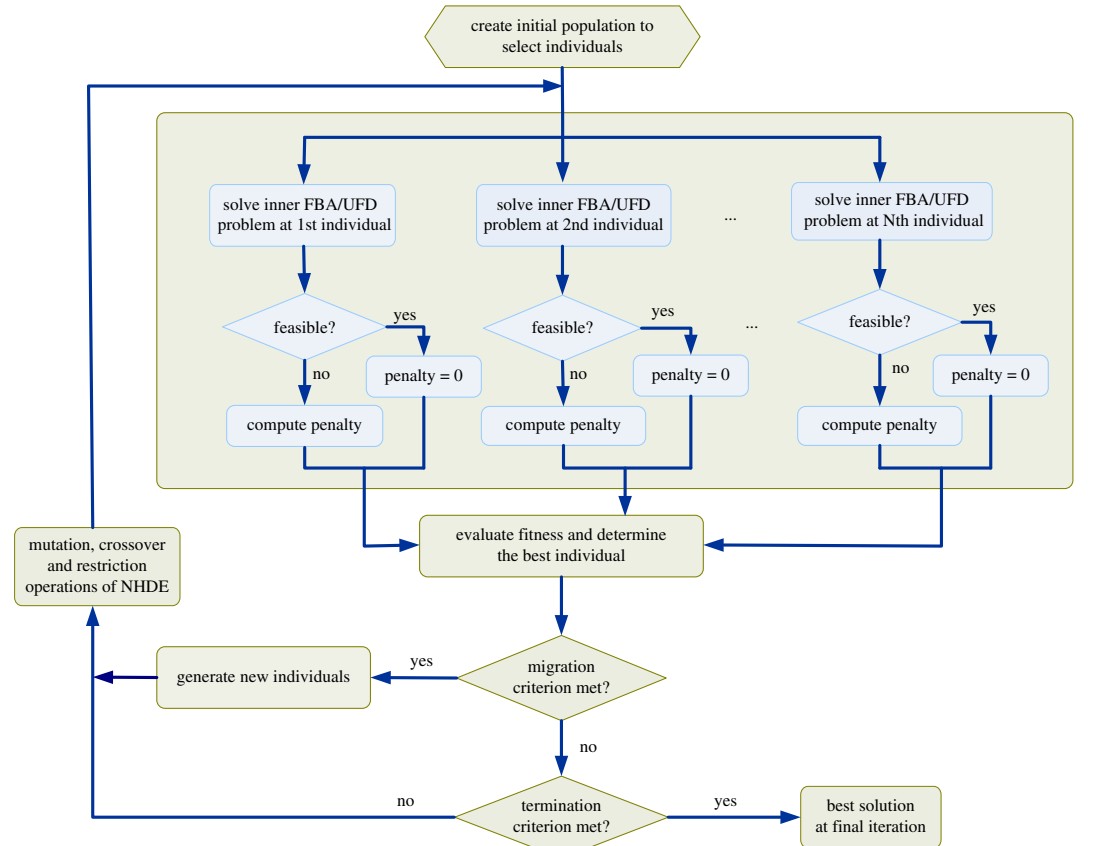

**Figure 3.** Flowchart of the NHDE algorithm. Hybrid differential evolution was applied to determine the outer decision variables and objective function. The inner optimization problem is a parallel computation in which each FBA problem was solved through linear programming to evaluate fitness in the NHDE algorithm.

# 3. Results and discussion

HNSCC, which accounts for more than 80% of head and neck malignancies, is one of the most common cancers worldwide. HNSCC is characterized by a high locoregional recurrence rate and includes cancers of the oral cavity, oropharynx, hypopharynx and larynx. The incidence and mortality rates of HNSCC vary by geographical location and site. A high incidence of oral cancer (excluding lip cancer) is found in South and Southeast Asia, parts of Western and Eastern Europe, parts of Latin America, and Caribbean and Pacific regions. Taiwan has one of the world's highest incidence rates of oral cancers. Improving the treatment outcome by identifying predictive targets for HNSCC treatment is an unmet clinical need. The aim of this study was to develop a computational strategy incorporated with expression profiles of proteins in head and neck tissue at normal and cancer states to reconstruct GSMN models.

## 3.1. Templates of flux-sum alterations

Recon 2.2 is a curated version of Recon 2 and can be used to ensure full elemental balancing, in turn facilitating constraint-based analyses [17]. It comprises 5324 metabolites, 7785 reactions and 1675 associated genes. To date, Recon 2.2 has not been employed to reconstruct a GSMN model of a tissue-specific cell. The proposed procedures presented in figure 1 were used to reconstruct GSMN models of HNSCs at healthy and cancer states. Total 12 865 gene expression profiles for human HNSCs and normal tissue were obtained from the HPA database. Based on the gene expression data and gene-protein-reaction (GPR) association, 384 and 411 high-confidence reactions at normal and cancer states were obtained, respectively, and applied to the CORDA algorithm to reconstruct the HNSC models. The reconstructed HNSC models comprised 2256 metabolites and 3427 reactions for the healthy (HT) model and 2147 metabolites and 3253 reactions for the cancer (CA) model. Both models identically had 82.2% and 75.7% of metabolites and reactions. The two models were combined to form a basal

**Table 1.** The inferred one-hit and two-hit oncogenes. AOX1, aldehyde oxidase; GFPT1, glutamine-fructose-6-phosphate aminotransferase [isomerizing] 1; GNPDA1, glucosamine-6-phosphate isomerase 1; IREB2, iron-responsive element-binding protein 2; OPLAH, 5-oxoprolinase; PGAM1, phosphoglycerate mutase 1; PI4KA, phosphatidylinositol 4-kinase alpha; PIK3CA, phosphatidylinositol 4,5-bisphosphate 3-kinase catalytic subunit alpha isoform; PIKFYVE, 1-phosphatidylinositol 3-phosphate 5-kinase; PIP5K1, phosphatidylinositol 4-phosphate 5-kinase type-1; RFK, riboflavin kinase; THTPA, thiamine-triphosphatase; TPK1, thiamin pyrophosphokinase 1.

| gene | metabolic subsystem | Ave. SR[a] | remark[b] |
| --- | --- | --- | --- |
| TPK1 | thiamine metabolism | 0.843 | vitamin B1/thiamine metabolism [31,32] |
| G6PC | glycolysis metabolism | 0.841 | metabolic regulator of glioblastoma [33] |
| GNPDA1 | aminosugar metabolism | 0.841 | cell migration [34] |
| PGAM1 | glycolysis metabolism | 0.841 | prognosis of OSCC/cell migration [35] |
| GFPT1 | aminosugar metabolism | 0.839 | prognosis in patients with pancreatic cancer [36] |
| RFK | vitamin B2 metabolism | 0.839 | ROS production/necrotic cell death [37] |
| THTPA | thiamine metabolism | 0.839 | invasion/metastasis of cancers [38,39] |
| OPLAH | glutathione metabolism | 0.838 | potential marker in some human cancers [40] |
| PTEN | inositol phosphate metabolism | 0.838 | tumour suppressor [41,42] |
| PI4KA | inositol phosphate metabolism | 0.837 | therapy of HCC [43] |
| PIKFYVE | glycerophospholipid metabolism | 0.837 | cancer cell migration/invasion [44] |
| PIP5K1 | glycerophospholipid metabolism | 0.836 | prognosis in prostate cancer [45] |
| PIK3CA | inositol phosphate metabolism | 0.835 | induce invasion in HNSCC [46] |
| (IREB2, AOX1) | citric acid cycle and vitamin B6 metabolism | 0.871 | biochemical recurrence of prostate cancer [47] |
| (RFK, IREB2) | vitamin B2 metabolism and citric acid cycle | 0.867 | see remark for RFK and IREB2 |
| (PTEN, IREB2) | inositol phosphate metabolism and citric acid cycle | 0.865 | cellular proliferation [48] |
| (G6PC, TPK1) | glycolysis and thiamine metabolism | 0.863 | see remark for G6PC and TPK1 |
| (G6PC, PGAM1) | glycolysis metabolism | 0.857 | see remark for G6PC and PGAM1 |

[a]Average similarity ratio of the mutant flux pattern to the template.
[b]Brief description of gene function and references.

(BL) model for investigating how a healthy cell can smoothly detour metabolic reprogramming to that of a cancer cell. The BL model was thus a union set of the HT and CA models and included 2417 metabolites and 3803 reactions. The three models were first employed in FBA and UFD problems to compute flux-sum distributions. We next used $LFC_m$ to compute the templates of flux-sum alterations for CA to BL and HT models.

## 3.2. Detecting one-hit oncogene

The NHDE algorithm was applied to solve the TLOP, and 13 of the top-ranked one-hit oncogenes were determined; their similarity ratios are shown in table 1. These enzymes participated in seven metabolic subsystems, namely inositol phosphate, glycerophospholipid, glycolysis, thiamine, amino sugar, glutathione and vitamin B2 metabolisms. Four genes (PTEN, PIKFYVE, PIK3CA and PI4KA) were detected in inositol phosphate metabolism, wherein the participating molecules were involved in cellular growth and proliferation signalling processes. These genes were highly connected with AKT/TP53/mTOR signalling pathway. Phosphatidylinositol 3,4,5-trisphosphate 3-phosphatase (PTEN), encoded by PTEN, was dysregulated, which yielded the maximum similarity ratios of $SR^{W,BL} = 0.9286$ and $SR^{W,HT} = 0.8929$ (electronic supplementary material, S2) for the Warburg effect related to the

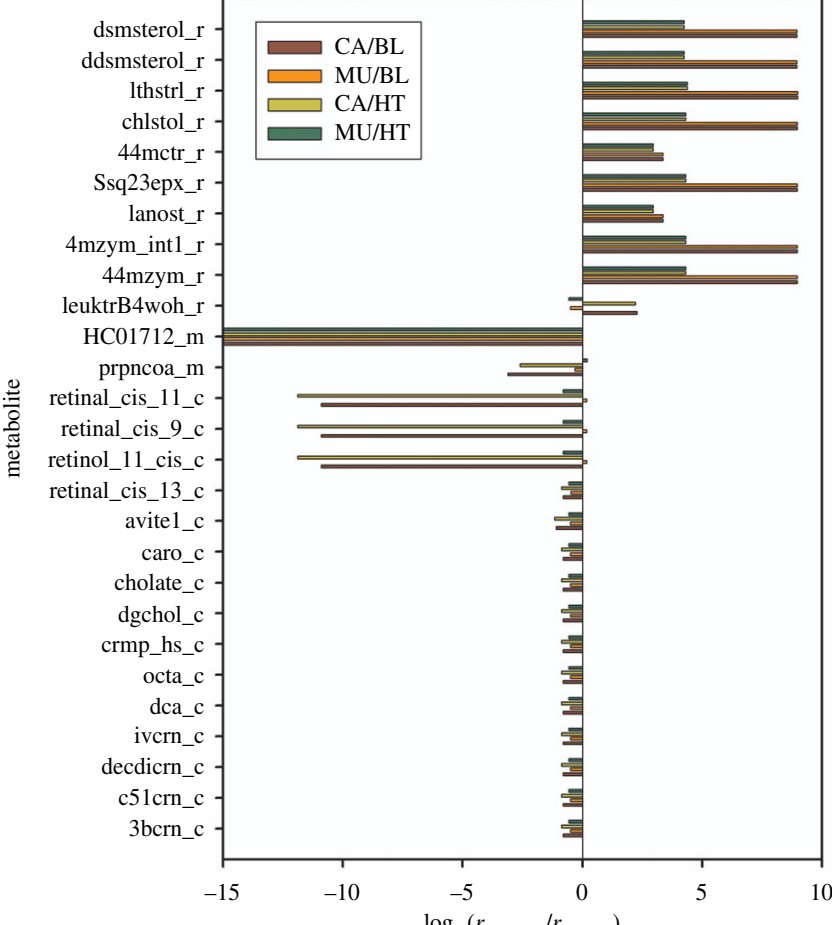

**Figure 4.** Logarithmic fold changes in the choke-point metabolites in lipid metabolism. In the endoplasmic reticulum, the levels of five cholesterols and their derivatives and four prenol lipids increased. Fifteen choke-point lipids were located in the cytoplasm, and their synthesis rates decreased, except for three retinoids. The synthesis rates of the cancer template and PTEN dysregulation for S(8)-glutaryl dihydrolipoamide (HC01712_m) in the mitochondria reduced to zero, thereby significantly reducing the LFC.

BL and HT models, respectively. *PTEN* dysregulation yielded the maximum ratios of $SR^{T,BL} = 0.7478$ and $SR^{T,HT} = 0.783$ compared with the templates of flux-sum alterations for the CA model to BL and HT models, respectively. The NHDE algorithm yielded the optimal average similarity (Ave. SR) ratio of 0.8381. *PTEN* is one of the most frequently observed tumour suppressor genes in human cancer. This biological cognition supports above computational results. Because altered AKT/PI3K/mTOR signalling has been frequently reported to contribute to human disease, researchers have made efforts to develop small-molecule inhibitors for AKT, PI3K and mTOR. Although the central role of PTEN in glucose metabolism has been well defined [31,43,46], little is known about how PTEN affects lipid metabolism [32,38]. The following results show the metabolic alterations by *PTEN* dysregulation.

Figure 4 shows the flux-sum alterations for 27 choke-point metabolites in lipid metabolism obtained using dysregulated PTEN. These choke-point metabolites were compartmented in the endoplasmic reticulum, mitochondrion and cytoplasm. In the endoplasmic reticulum, the flux-sum (LFC > 0) of five cholesterols and their derivatives and four prenol lipids increased because of *PTEN* dysregulation. This observation indicated that the metabolite level should increase. The metabolite 20-hydroxy-leukotriene B4 (leuktrB4woh_r) was classified as a fatty acyl in the endoplasmic reticulum; its synthesis rates in the templates increased but was inconsistent in dysregulated cases, as shown in figure 4. The synthesis rates of S(8)-glutaryl dihydrolipoamide (HC01712_m) in the mitochondria in the cancer template and *PTEN* dysregulated cases reduced to zero; thus, the LFC was significantly reduced. The LFC of the template for acryloyl-CoA (prpncoa_m) was inconsistent with the dysregulation. Fifteen choke-point lipids were located in the cytoplasm, and their synthesis rates decreased, except for three retinoids that exhibited increased flux rates for dysregulation. *N*-acylsphingosine 1-phosphate (crmp_hs_c) was classified as a sphingolipid in the cytoplasm, and its synthesis rate in the dysregulated cases was consistent with that in the template. Furthermore, the

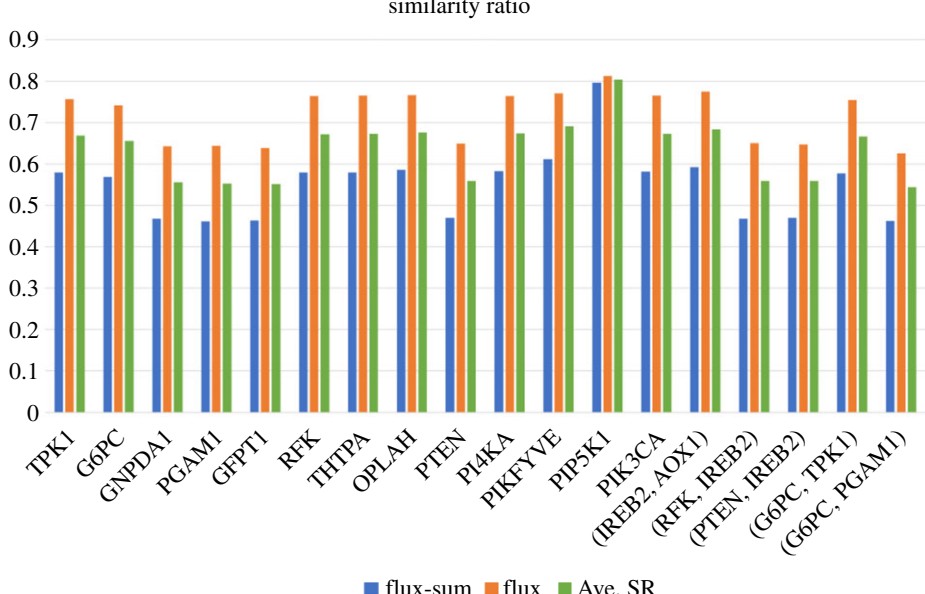

**Figure 5.** Similarity ratios of flux and flux-sum synthesis rates evaluated by flux variability analysis in a posterior inspection. The similarity ratio for each mutant is the ratio of the number of metabolites with same flux/flux-sum trends in the flux interval of the mutant to those of the template.

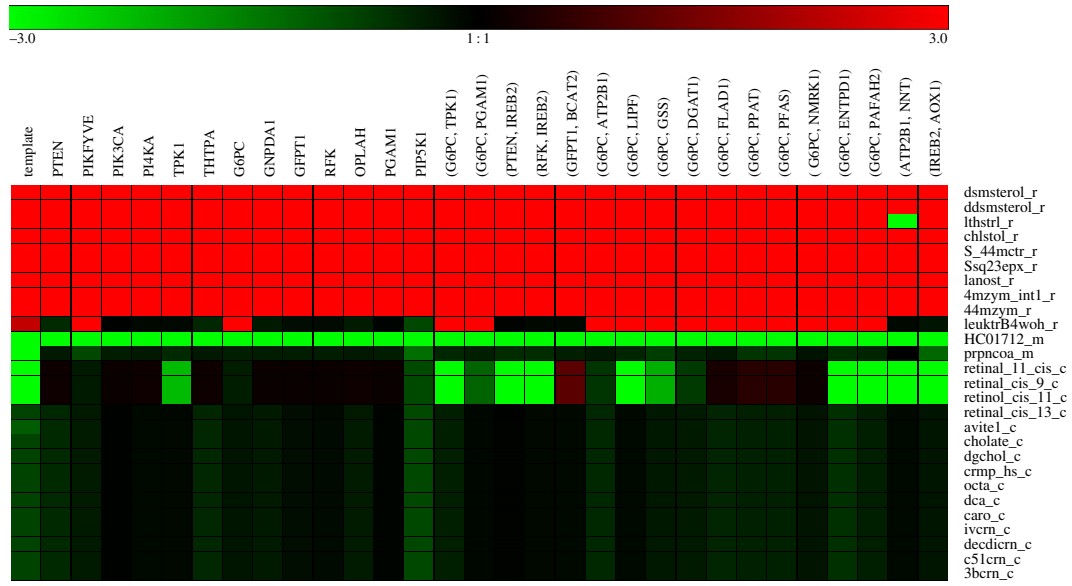

**Figure 6.** Hierarchical clustering of logarithmic fold changes in the choke-point metabolites. Hierarchical clustering of logarithmic fold changes, $\log_2 (r_{\text{deficient}}/r_{\text{normal}})$, in the choke-point metabolites in lipid metabolism for the template and all one-hit and two-hit deficiencies. Red colour indicates an increase, and green indicates a decrease.

computational results that revealed increased cholesterol levels and decreased sphingolipid levels are consistent with the results for various carcinomas in the literature [49–51].

Genes from the *PIKFYVE*, *PIK3CA* and *PI4KA* families were also detected in inositol phosphate metabolism. In this study, the maximum similarity ratios of $SR^{W,BL} = 0.9286$, $SR^{W,HT} = 0.8929$, $SR^{T,BL} = 0.7424$ and $SR^{T,HT} = 0.783$ were yielded through computation in the dysregulated cases. PIKFYVE has been reported to mediate EGFR trafficking to the nucleus in human bladder cancer tissue [52] and regulate lung carcinoma cell migration and invasion through RAC1 activation [44]. The similarity ratios for PIK3CA and PI4KA were nearly equal to those for PTEN. Dysregulation of PIK3CA and PI4KA is commonly found in HNSCC [46] and hepatocellular carcinoma [43], respectively.

Nine of the top-ranked one-hit oncogenes (table 1) had identical similarity ratios for Warburg effects, except for glucose-6-phosphatase (G6PC), which had a slightly higher ratio ($SR^{W,HT} = 0.8408$) than the

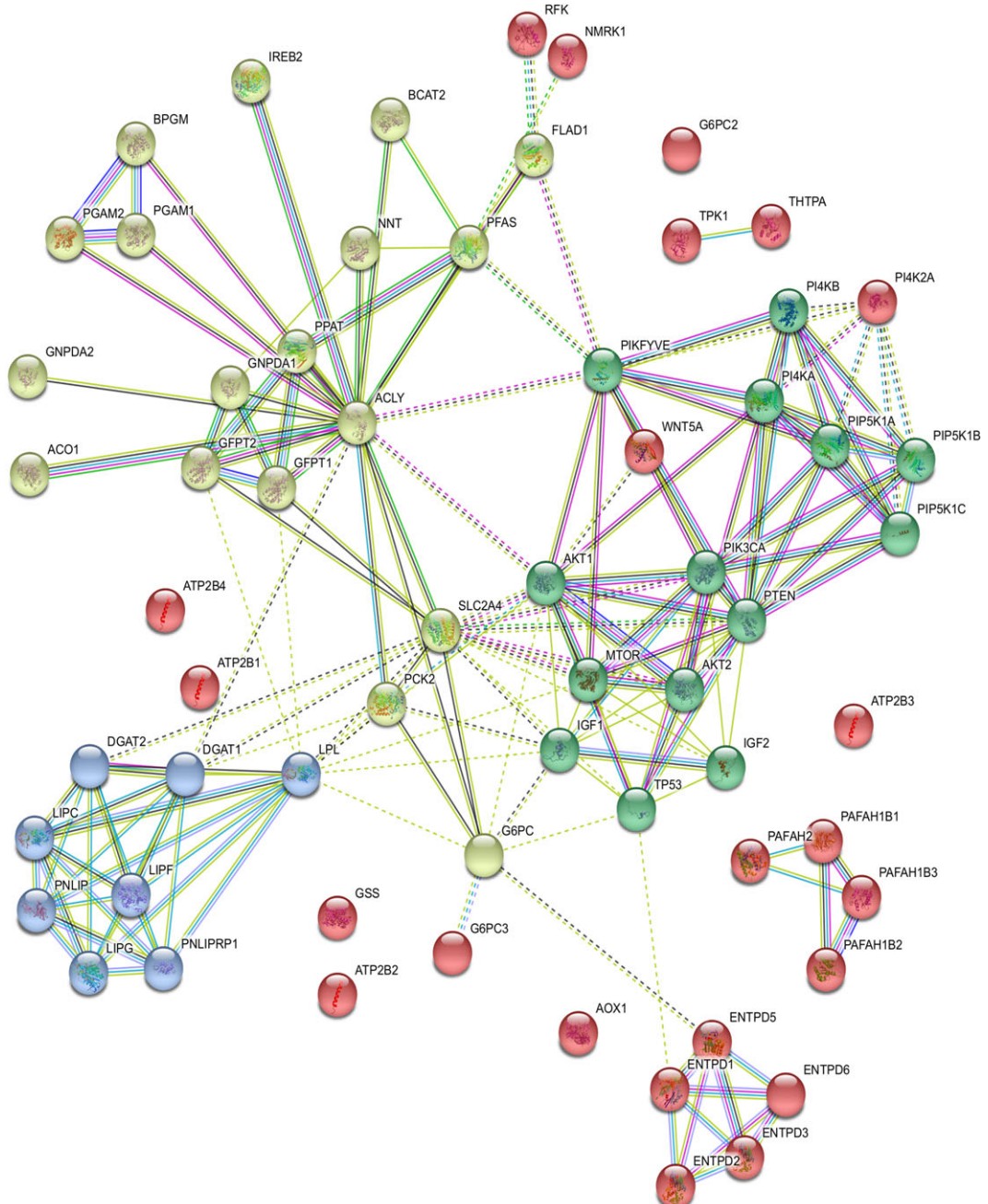

**Figure 7.** Protein–protein interactions of inferred oncogenes. Protein–protein interactions (PPIs) of the inferred oncogenes with eight additional signalling proteins and their families, namely ACLY, TP53, AKT, IGF, GSK, GLUT, PEPCK and MTOR, *k*-means clustering function in STRING database was applied to classify the PPI network into four classes denoted by four colours. The proteins in the first cluster (green balls) participated in inositol phosphate and glycerophospholipid metabolism, related to PTEN. The proteins in the second cluster (emerald green balls) were involved in triacylglycerol synthesis. The proteins in the third cluster (purple balls) were mainly involved in gluconeogenesis and amino sugar and citric acid cycle metabolisms. The proteins in the fourth cluster (red balls) were involved in other diversified metabolisms.

ratios of the other oncogenes. However, the similarity ratios of the templates were slightly different from those of PTEN. According to a survey of PubMed and cancer databases, each dysregulated enzyme could cause various cancers and diseases, as shown in table 1 and electronic supplementary material, S2.

Evidence indicates that cancer is the result of accumulated genetic mutations, also known as the Knudson hypothesis [53]. Following the same procedures (described in above section), the NHDE algorithm was applied to detect two-hit enzyme deficiencies. A total of 17 of the top-ranked two-hit enzymes yielded higher similarity ratios than those of one-hit enzymes (table 1; electronic supplementary

material, S2). They could achieve the optimal Ave. SRs about 0.86, which were approximately 3% higher than those for the one-hit dysregulation. A survival analysis for the oncogenes can be applied to investigate the clinical significance of the metabolic alterations. However, such investigations need several clinical trials. In this study, we surveyed survival analysis from the HPA to explain survival significance of the inferred oncogenes, and the detailed results were also shown in electronic supplementary material, S3. We observed that five genes (*GNPDA1*, *PGAM1*, *GFPT1*, *RFK* and *PTEN*) are significant for survival.

## 3.3. Flux variability analysis

The FBA in the TLOP problem was employed to evaluate flux alterations for normal and cancer model. However, it is a bias method to compute the flux pattern. Flux variability analysis (FVA) can be applied in a posterior inspection to overcome such a drawback, but required a lot of computation time. We applied the FVA to compute the minimum and maximum quantities of each metabolite for the normal model and the mutants, respectively. The flux intervals of each deficient case were compared with the normal intervals, and classified into seven categories by using the definition presented in the electronic supplementary material, S1. We evaluated the trends of flux and flux-sum synthesis rate for each metabolite of the mutants and those of the template, and then counted the number of metabolites with same flux/flux-sum trends in the flux interval for each mutant and the template to obtain similarity ratios as shown in figure 5. We found that the flux-sum similarity ratios and flux similarity ratios of all mutants are greater than 47% and 65%, respectively, so that the average ratio for each mutant is higher than 56%.

## 3.4. Connection with signal pathway

We also investigated behaviours of lipid metabolism due to one-hit and two-hit deficiencies obtained from table 1. The LFCs of the choke-point metabolites for the templates and mutants are shown in figure 6 and electronic supplementary material, S4. For all dysregulations, we observed that the levels of five cholesterols and four prenol lipids in the endoplasmic reticulum increased, except for $5\alpha$-cholest-7-en-3$\beta$-ol (lthstrl_r) dysregulated by the (ATP2B1, NNT) pair. Furthermore, sphingolipid (crmp_hs_c) and HC01712_m levels decreased; the same results were obtained for PTEN. In addition, the synthesis rates of all dysregulations of HC01712_m reduced to zero, which were consistent with those for the templates. The choke-point metabolites could be used as potential biomarker candidates for detecting the metabolic programming between cancer and normal states.

We used the STRING database [54] to investigate protein–protein interactions (PPIs) for the inferred oncogenes obtained from table 1. Eight signalling proteins and their families, namely ACLY, TP53, AKT, IGF, GSK, GLUT, PEPCK and MTOR, were included in the survey and were related to cellular quiescence, proliferation, cancer and longevity in an intracellular signalling pathway. Almost all the proteins were strongly connected, and only a few others independently existed in the PPI network. We used *k*-means clustering to classify the network into four clusters (see figure 7 and electronic supplementary material S5 for details). The proteins in the first cluster (green balls) participated in inositol phosphate and glycerophospholipid metabolisms and were linked with the TP53/AKT/IGF/ MTOR signalling pathway through PTEN. The proteins in the second cluster (emerald green balls) were involved in triacylglycerol synthesis and linked with PIKFYVE and AKT1 in the first cluster through ACLY. The proteins in the third cluster (purple balls) were mainly involved in gluconeogenesis and amino sugar and citric acid cycle metabolisms. It also connected with the second cluster. The proteins in the last cluster (red balls) were involved in other diversified metabolisms.

# 4. Conclusion

A tri-level inference optimization framework was applied to detect 13 one-hit and 17 two-hit deficient enzymes that contribute to inducing HNSCC. According to the PubMed survey, each detected one-hit deficiency participated in carcinogenesis in various tissues. The inferred oncogene *PTEN* is one of the most frequently mutated tumour suppressor genes in human cancer. In recent years, the function of PTEN as a metabolic regulator has attracted considerable research attention. This study also supported this function. From the literature survey, we found that *TPK1* is an oncogene for colon cancer, and three genes (*PTEN*, *GNPDA1* and *PIK3CA*) appear in head and neck cancer. Deficiency of G6PC leads to accelerated hepatic carcinogenesis in glycogen storage disease. In addition, G6PC was involved in 11 pairs of two-hit oncogenes, as revealed by the computation in this study. Moreover, we

will conduct wet-laboratory experiment to verify the inferred results in order to discover potential cancer cell metabolic targets.

Data accessibility. The datasets supporting this article have been uploaded as part of the electronic supplementary material. The optimization code used during this study is available via the Dryad Digital Repository https://doi.org/10.5061/dryad.364vk23 (doi:10.5061/dryad.364vk23) [55].

Authors' contributions. W.-H.W. wrote the modelling software and revised the manuscript; F.-Y.L. and Y.-C.S. collected field data and carried out the analyses and paper survey; J.-M.L., P.M.-H.C. and C.-Y.F.H. participated in the design of the study and data analysis; F.-S.W. conceived of the study, designed the study, coordinated the study and drafted the manuscript. All authors gave final approval for publication and agree to be held accountable for the work performed therein.

Competing interests. The authors declare that they have no competing interests.

Funding. This work was supported by Ministry of Science and Technology of Taiwan (MOST) (grant nos. MOST106-2221-E-194-049-MY3 and MOST107-2627-M-194-001 to F.-S.W., MOST107-2627-M-030-001 to J.-M.L., MOST107-2627-M-075-001 to P.M.-H.C., MOST107-2320-B-010-040-MY3 to C.-Y.F.H.).

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
