## [Reviewer comments · Royal Society Open Science]

Review History

RSOS-191241.R0 (Original submission)

Review form: Reviewer 1

Is the manuscript scientifically sound in its present form?

No

Are the interpretations and conclusions justified by the results?

Yes

Is the language acceptable?

Yes

Do you have any ethical concerns with this paper?

No

Have you any concerns about statistical analyses in this paper?

No

Recommendation?

Major revision is needed (please make suggestions in comments)

Comments to the Author(s)

The manuscript by Wu et al. introduces a trilevel optimisation problem to simulate cancer and healthy genome-scale metabolic networks, with a case study on head and neck squamous cells.

Major points

The manuscript claims novelty in a few points, but in my opinion does not justify why the approach is novel. In general, throughout the manuscript, the steps are not well explained. The reader is shown some results and conclusions, and is left trying to reverse engineer the approach used by the authors. The methods used and the variables assigned to each algorithm should be carefully listed.

The authors claim the trilevel approach is novel in this context. However, a trilevel formulation has been previously proposed in cancer metabolic modelling, e.g. doi: 10.1093/bioinformatics/btx562. The authors should revise their claim or contextualise and discuss the differences within previous approaches.

The authors also claim that this is the first time that tissue-specific models are built from Recon 2.2. This claim is not accurate, see e.g. doi: 10.1371/journal.pcbi.1006867, doi:10.1073/pnas.1713050114. Again, the authors should revisit/contextualise their claim acknowledging previous work.

The way in which flux-sum is used to obtain metabolite information is not clear and should be made more explicit. Have the authors used the default flux-sum approach? Is this the one outlined in Figure 6?

Also, the mathematical formulation and the results do not transmit confidence in the proposed approach. For instance, Fig. 5 has a metabolite where the tiny ratio is likely to come from a solver error, or a problem with the tolerance, but the authors did not spot or correct this. Also, Figure 6 seems a clustering made on a single value from Figure 5, which is affected by the above problem.

On the outer and inner problems, the authors should clarify a major point: are the UFD problem and the FBA problem solved at the same level? If so, then this is actually a bilevel problem, and repeating the constraints $NV=0$ and $vLB < v < vUB$ is redundant. If not, then why are they not nested in the mathematical formulation? I can assume that the UFD is run as an outer problem compared to the FBA? Otherwise, how is the flux distribution calculated for the UFD? In any case, all these strategies (including norm-2 regularisation) have been adopted in the past in metabolic modelling, and the authors should clearly identify their contribution.

The selection of the objective function is not clear. Are the authors optimising ATP and biomass with equal weight? Or is the objective selected according to the type of cancer/healthy model? If so, the authors should demonstrate that the flux distributions are comparable even when different objective functions are considered, and the results are not just driven by the change in the objective function. E.g., can the authors show or discuss that the effect of changing the objective function when comparing flux or flux-sum distributions in healthy-vs-cancer is negligible compared to the actual difference of flux in healthy-vs-cancer?

The authors introduce similarity ratios. They should justify why this further discretisation of $u_m^{(T/W)}$ is needed, as it introduces another parameter (the tolerance). Is this discretisation the reason why the authors need to introduce the TLOP to solve the discretised problem? Could this be left as a continuous variable, avoiding the NHDE heuristic altogether? Why the NHDE is needed exactly, and how is it formulated (specifically, how are individuals defined?).

Minor points

The manuscript has several typos, most of which could have been easily fixed with careful proofreading. E.g. "according literature survey", "than do normal cells"

And infer that -> "and inferred that

Used the CORDA -> used CORDA

Figure 4 seems created with STRING, and the authors should mention it in the caption. Also, they refer to clusters as "first cluster", "second cluster" etc., but the reader has no way to check the order of clusters from the figure.

Review form: Reviewer 2 (Sriram Chandrasekaran)

Is the manuscript scientifically sound in its present form?

Yes

Are the interpretations and conclusions justified by the results?

No

Is the language acceptable?

Yes

Do you have any ethical concerns with this paper?

No

Have you any concerns about statistical analyses in this paper?

Yes

Recommendation?

Major revision is needed (please make suggestions in comments)

Comments to the Author(s)

This is an interesting study to find metabolic oncogenes. The authors use a genome scale metabolic modeling approach to identify metabolic genes that when perturbed rewires normal cell metabolism towards cancer state. This is achieved by using a template cancer state and comparing the similarity of the metabolic state after gene knockdowns with the template. The approach is novel and useful for the cancer metabolism community.

Major revisions

One major concern is that the authors use one specific type of cancer (head and neck). But the driver genes they identify (e.g. PTEN) are not specific to that cancer. Many of the oncogenes are observed in other cancers and its unclear if they are unique to head and neck. I'd suggest that the authors repeat the analysis in 2 or three other cancer types to identify to those genes that are specific to a cancer type and those that are more general. Alternately, they should compare their predicted oncogenes with experimental data specific to head and neck cancer alone.

Most cancers have subtypes with distinct genomic features, including head and neck cancer. Can this data be stratified in to those sub types? Do those sub types have distinct metabolic activity? By lumping the sub types they may be mixing up signal from different sub types.

The basal model doesnt make biological sense to me. The authors should have either used the entire recon2 model or started with the normal model and identified interventions that make it

similar to cancer state. I can understand why they did it from a computational perspective but the biological significance of the basal model is unclear.

The authors haven't rigorously pursued the clinical significance of the metabolic alterations. Does downregulation or mutation in these enzymes lead to reduced survival? If these genes are not associated with patient survival, then they may not be oncogenic drivers.

There's no statistical validation of the predictions. There are numerous metabolic oncogenes. So it is unclear if the genes identified by the authors could have been arrived at by random chance. They should do a sampling analysis (of metabolic genes in the model) and identify overlap with all known metabolic oncogenes. The overlap should then be compared with the overlap obtained by their approach.

-Sriram Chandrasekaran

Review form: Reviewer 3

Is the manuscript scientifically sound in its present form?

Yes

Are the interpretations and conclusions justified by the results?

Yes

Is the language acceptable?

Yes

Do you have any ethical concerns with this paper?

No

Have you any concerns about statistical analyses in this paper?

No

Recommendation?

Accept with minor revision (please list in comments)

Comments to the Author(s)

In this work, the authors utilized Recon2.2 (a human metabolic network reconstruction) and the Human Protein Atlas database in order to reconstruct a head and neck squamous cell-specific network describing both the healthy and the cancer cells. They used the Cost Optimization Reaction Dependency Assessment (CORDA) algorithm proposed by Schultz and Qutub to build the genome-scale models in a tissue-specific manner. On top of that, they developed a tri-level optimization framework to identify dysregulated enzymes in this cancer type. This is an interesting work that deserves to be published. However, there are a number of minor issues that would need to be addressed before the manuscript can be suitable for publication.

Overall, there are too many acronyms in the manuscript, which make it hard to follow.

Furthermore, as might not all readers be familiar with the terminology used in metabolic models it would be nice if some definitions are given i.e. choke-point metabolites, flux-sum analysis, as well as other terms used in this manuscript like templates, mutant flux patterns, etc.

It would be nice if, at the end of the introduction where the current work is described, you could add more the intuition between the algorithms used instead of repeating the algorithmic steps.

Furthermore, the tri-level optimization algorithm in Methods should be better explained. Maybe you could give the overall idea by describing each optimization problem and then become more mathematically specific. I was always come back and forth in order to understand it.

More specifically,

Introduction:

-Please rephrase: "Flux-dependent and pruning methods are automated tissue-specific reconstruction algorithms for generating a genome-scale metabolic model for predicting tissue-specific behaviours [6, 10]."

- A tri-level inference optimization framework: Say here what it does in a few words... Please, give the underlying intuition.

- HNSCC: I think this is redundant considering that you have already defined HNSCs.

Methods:

- HNSC: use the acronym instead.

- definition or reference of flux-sum alterations

- "The basal model (union set of CA and HT models) was used": better move the definition at the beginning where you first mention it.

- "TLOP was used to simulate a wet lab..." what exactly do you mean by that?

- chokepoint metabolites: give the definition

- "The objective function in the outer optimization problem is a similarity measure that..." Please make clearer what the outer objective does. Give the intuition behind.

- "the Warburg hypothesis obtained from the literature..." What exactly do you obtain from the literature? Please, be more precise. Please, add the related references.

- Shouldn't the definition of SR go first together with its meaning in order to understand the trilevel optimization?

- the logarithmic fold changes: Add the acronym LFC where you first mention it. Not later.

- UFD is defined later on page 13.

Results and Discussion

-section 3.1: shouldn't this section be presented earlier in the manuscript in methods... it helps a lot the overall understanding.

Decision letter (RSOS-191241.R0)

17-Jan-2020

Dear Miss Wang,

The editors assigned to your paper ("Oncogene inference optimization using constraint-based modelling incorporated with protein expression in normal and tumour tissues") have now received comments from reviewers. We would like you to revise your paper in accordance with the referee and Associate Editor suggestions which can be found below (not including confidential reports to the Editor). Please note this decision does not guarantee eventual acceptance.

Please submit a copy of your revised paper before 09-Feb-2020. Please note that the revision deadline will expire at 00.00am on this date. If we do not hear from you within this time then it will be assumed that the paper has been withdrawn. In exceptional circumstances, extensions may be possible if agreed with the Editorial Office in advance. We do not allow multiple rounds of revision so we urge you to make every effort to fully address all of the comments at this stage. If deemed necessary by the Editors, your manuscript will be sent back to one or more of the original reviewers for assessment. If the original reviewers are not available, we may invite new reviewers.

To revise your manuscript, log into <http://mc.manuscriptcentral.com/rsos> and enter your Author Centre, where you will find your manuscript title listed under "Manuscripts with Decisions." Under "Actions," click on "Create a Revision." Your manuscript number has been

appended to denote a revision. Revise your manuscript and upload a new version through your Author Centre.

- Data accessibility

If you wish to submit your supporting data or code to Dryad (<http://datadryad.org/>), or modify your current submission to dryad, please use the following link:
<http://datadryad.org/submit?journalID=RSOS&manu=RSOS-191241>

- Competing interests

- Authors' contributions

- Acknowledgements

- Funding statement

Kind regards,

Andrew Dunn

on behalf of Dr Simon Cook (Associate Editor) and Pietro Cicuta (Subject Editor)

Editor comments:

Please address in detail the points raised by reviewers.

Comments to Author:

Reviewers' Comments to Author:

Reviewer: 1

Comments to the Author(s)

The manuscript by Wu et al. introduces a trilevel optimisation problem to simulate cancer and healthy genome-scale metabolic networks, with a case study on head and neck squamous cells.

Major points

The manuscript claims novelty in a few points, but in my opinion does not justify why the approach is novel. In general, throughout the manuscript, the steps are not well explained. The reader is shown some results and conclusions, and is left trying to reverse engineer the approach used by the authors. The methods used and the variables assigned to each algorithm should be carefully listed.

The authors claim the trilevel approach is novel in this context. However, a trilevel formulation has been previously proposed in cancer metabolic modelling, e.g. doi: 10.1093/bioinformatics/btx562. The authors should revise their claim or contextualise and discuss the differences within previous approaches.

The authors also claim that this is the first time that tissue-specific models are built from Recon 2.2. This claim is not accurate, see e.g. doi: 10.1371/journal.pcbi.1006867, doi:10.1073/pnas.1713050114. Again, the authors should revisit/contextualise their claim acknowledging previous work.

The way in which flux-sum is used to obtain metabolite information is not clear and should be made more explicit. Have the authors used the default flux-sum approach? Is this the one outlined in Figure 6?

Also, the mathematical formulation and the results do not transmit confidence in the proposed approach. For instance, Fig. 5 has a metabolite where the tiny ratio is likely to come from a solver

error, or a problem with the tolerance, but the authors did not spot or correct this. Also, Figure 6 seems a clustering made on a single value from Figure 5, which is affected by the above problem.

On the outer and inner problems, the authors should clarify a major point: are the UFD problem and the FBA problem solved at the same level? If so, then this is actually a bilevel problem, and repeating the constraints $NV=0$ and $vLB < v < vUB$ is redundant. If not, then why are they not nested in the mathematical formulation? I can assume that the UFD is run as an outer problem compared to the FBA? Otherwise, how is the flux distribution calculated for the UFD? In any case, all these strategies (including norm-2 regularisation) have been adopted in the past in metabolic modelling, and the authors should clearly identify their contribution.

The selection of the objective function is not clear. Are the authors optimising ATP and biomass with equal weight? Or is the objective selected according to the type of cancer/healthy model? If so, the authors should demonstrate that the flux distributions are comparable even when different objective functions are considered, and the results are not just driven by the change in the objective function. E.g., can the authors show or discuss that the effect of changing the objective function when comparing flux or flux-sum distributions in healthy-vs-cancer is negligible compared to the actual difference of flux in healthy-vs-cancer?

The authors introduce similarity ratios. They should justify why this further discretisation of $u_m^{(T/W)}$ is needed, as it introduces another parameter (the tolerance). Is this discretisation the reason why the authors need to introduce the TLOP to solve the discretised problem? Could this be left as a continuous variable, avoiding the NHDE heuristic altogether? Why the NHDE is needed exactly, and how is it formulated (specifically, how are individuals defined?).

Minor points

The manuscript has several typos, most of which could have been easily fixed with careful proofreading. E.g. "according literature survey", "than do normal cells"

And infer that -> "and inferred that"

Used the CORDA -> used CORDA

Figure 4 seems created with STRING, and the authors should mention it in the caption. Also, they refer to clusters as "first cluster", "second cluster" etc., but the reader has no way to check the order of clusters from the figure.

Reviewer: 2

Comments to the Author(s)

This is an interesting study to find metabolic oncogenes. The authors use a genome scale metabolic modeling approach to identify metabolic genes that when perturbed rewires normal cell metabolism towards cancer state. This is achieved by using a template cancer state and comparing the similarity of the metabolic state after gene knockdowns with the template. The approach is novel and useful for the cancer metabolism community.

Major revisions

One major concern is that the authors use one specific type of cancer (head and neck). But the driver genes they identify (e.g. PTEN) are not specific to that cancer. Many of the oncogenes are observed in other cancers and its unclear if they are unique to head and neck. I'd suggest that the authors repeat the analysis in 2 or three other cancer types to identify to those genes that are specific to a cancer type and those that are more general. Alternately, they should compare their predicted oncogenes with experimental data specific to head and neck cancer alone.

Most cancers have subtypes with distinct genomic features, including head and neck cancer. Can

this data be stratified in to those sub types? Do those sub types have distinct metabolic activity? By lumping the sub types they may be mixing up signal from different sub types.

The basal model doesnt make biological sense to me. The authors should have either used the entire recon2 model or started with the normal model and identified interventions that make it similar to cancer state. I can understand why they did it from a computational perspective but the biological significance of the basal model is unclear.

The authors havent rigorously pursued the clinical significance of the metabolic alterations. Does downregulation or mutation in these enzymes lead to reduced survival? If these genes are not associated with patient survival, then they may not be oncogenic drivers.

There's no statistical validation of the predictions. There are numerous metabolic oncogenes. So it is unclear if the genes identified by the authors could have been arrived at by random chance. They should do a sampling analysis (of metabolic genes in the model) and identify overlap with all known metabolic oncogenes. The overlap should then be compared with the overlap obtained by their approach.

-Sriram Chandrasekaran

Reviewer: 3

Comments to the Author(s)

In this work, the authors utilized Recon2.2 (a human metabolic network reconstruction) and the Human Protein Atlas database in order to reconstruct a head and neck squamous cell-specific network describing both the healthy and the cancer cells. They used the Cost Optimization Reaction Dependency Assessment (CORDA) algorithm proposed by Schultz and Qutub to build the genome-scale models in a tissue-specific manner. On top of that, they developed a tri-level optimization framework to identify dysregulated enzymes in this cancer type. This is an interesting work that deserves to be published. However, there are a number of minor issues that would need to be addressed before the manuscript can be suitable for publication.

Overall, there are too many acronyms in the manuscript, which make it hard to follow.

Furthermore, as might not all readers be familiar with the terminology used in metabolic models it would be nice if some definitions are given i.e. choke-point metabolites, flux-sum analysis, as well as other terms used in this manuscript like templates, mutant flux patterns, etc.

It would be nice if, at the end of the introduction where the current work is described, you could add more the intuition between the algorithms used instead of repeating the algorithmic steps.

Furthermore, the tri-level optimization algorithm in Methods should be better explained. Maybe you could give the overall idea by describing each optimization problem and then become more mathematically specific. I was always come back and forth in order to understand it.

More specifically,

Introduction:

-Please rephrase: "Flux-dependent and pruning methods are automated tissue-specific reconstruction algorithms for generating a genome-scale metabolic model for predicting tissue-specific behaviours [6, 10]."

- A tri-level inference optimization framework: Say here what it does in a few words... Please, give the underlying intuition.

- HNSCC: I think this is redundant considering that you have already defined HNSCs.

Methods:

- HNSC: use the acronym instead.

- definition or reference of flux-sum alterations

- "The basal model (union set of CA and HT models) was used": better move the definition at the beginning where you first mention it.

- "TLOP was used to simulate a wet lab..." what exactly do you mean by that?

- chokepoint metabolites: give the definition

- “The objective function in the outer optimization problem is a similarity measure that...” Please make clearer what the outer objective does. Give the intuition behind.
 - “ the Warburg hypothesis obtained from the literature...” What exactly do you obtain from the literature? Please, be more precise. Please, add the related references.
 - Shouldn't the definition of SR go first together with its meaning in order to understand the trilevel optimization?
 - the logarithmic fold changes: Add the acronym LFC where you first mention it. Not later.
 - UFD is defined later on page 13.
- Results and Discussion
- section 3.1: shouldn't this section be presented earlier in the manuscript in methods... it helps a lot the overall understanding.

Author's Response to Decision Letter for (RSOS-191241.R0)

See Appendix A.

Decision letter (RSOS-191241.R1)

26-Feb-2020

Dear Miss Wang,

It is a pleasure to accept your manuscript entitled "Oncogene inference optimization using constraint-based modelling incorporated with protein expression in normal and tumour tissues" in its current form for publication in Royal Society Open Science.

on behalf of Dr Simon Cook (Associate Editor) and Pietro Cicuta (Subject Editor)
openscience@royalsociety.org

Appendix A

Reviewers' Comments to Author:

Reviewer: 1

Comments to the Author(s)

The manuscript by Wu et al. introduces a trilevel optimisation problem to simulate cancer and healthy genome-scale metabolic networks, with a case study on head and neck squamous cells.

Thank you for the positive reviews and constructive criticism. You have spent a lot of time on this manuscript, and we appreciate their effort. We have carefully considered all recommendations and followed essentially all of them, as is detailed below.

Major points

The manuscript claims novelty in a few points, but in my opinion does not justify why the approach is novel. In general, throughout the manuscript, the steps are not well explained.

We have revised the manuscript and redrawn Figure 2 (as shown in below) to explain the design concept of the algorithm.

Fig 2. Oncogene inference problem incorporated with the templates.

The oncogene inference framework integrated the templates and the TLOP is used to mimic gene screening procedures in a wet lab to detect dysregulated oncogenes in head and neck squamous cells. The first, second and third procedures in Figure 2 are discussed in Figure 1 to build the flux template, which acts as a control for the

oncogene inference problem. GSMN models of cancerous and normal cells are reconstructed, as shown in Figure 2A. Both models are then applied to compute the flux distribution patterns at the cancer and normal situations (Figure 2B). The flux template is built according to the flux distributions of cancer and normal models (Figure 2C). Blue arrows (Figure 2D to J) present TLOP to mimic mutant schemes in a wet lab for oncogene inference. The outer optimization problem is employed to decide which genes are mutated and their corresponding dysregulated bounds. The decision variables are provided for FBA and UFD problems to hierarchically compute flux distribution for each mutant. The fluxes acted as the decision variables in FBA and UFD problems are then applied to evaluate the similarity measures in the outer problem. The evolutionary computation is repeatedly until to achieve the optimal result.

The reader is shown some results and conclusions, and is left trying to reverse engineer the approach used by the authors. The methods used and the variables assigned to each algorithm should be carefully listed.

We have explained the variables in the revised manuscript. The outer optimization problem is employed to decide mutated genes and their corresponding dysregulated bounds. The decision variables were used in the FBA and UFD problems to hierarchically compute flux distribution for each mutant. The fluxes obtained from the decision variables in FBA and UFD problems were then applied to evaluate the similarity ratios in the outer problem.

The authors claim the trilevel approach is novel in this context. However, a trilevel formulation has been previously proposed in cancer metabolic modelling, e.g. doi: 10.1093/bioinformatics/btx562. The authors should revise their claim or contextualise and discuss the differences within previous approaches.

We have cited and discussed the article in the Introduction section. A tri-level optimization problem (TLOP) integrating splice-isoform expression has been introduced to depict breast cancer metabolism [23]. This study introduced a similarity measure in the TLOP to decide mutated genes and their corresponding dysregulated bounds.

The authors also claim that this is the first time that tissue-specific models are built from Recon 2.2. This claim is not accurate, see e.g. doi: 10.1371/journal.pcbi.1006867, doi:10.1073/pnas.1713050114. Again, the authors should revisit/contextualise their claim acknowledging previous work.

We have cited and discussed the two articles in the Introduction section. Recently, Richelle *et al.* [18] built 44 different genome-scale metabolic models from Recon 2.2

[17] and iHsa [19] using six model extraction methods with RNA-Seq data from NCI-60 cell line. Such an approach provides guidelines for the development of the next-generation of data contextualization methods. Ryu *et al.* [20] presented a systematic framework for the generation of gene-transcript-protein-reaction associations in the human metabolism and addition of new reactions from Recon 2.2 to build Recon 2M.2 that is biochemically consistent and transcript-level data compatible. Such a gene-transcript-protein-reaction information enabled more accurate simulation of cancer metabolism and prediction of anticancer targets.

The way in which flux-sum is used to obtain metabolite information is not clear and should be made more explicit. Have the authors used the default flux-sum approach? Is this the one outlined in Figure 6?

We have provided a numerical example (supplementary material, S1) to illustrate definitions of flux-sum, logarithmic fold change ratio and similarity ratio. For example, suppose that a metabolic network consisted of the metabolite, M1, which participates in cytoplasm (c) and mitochondrion (m). Five reversible reactions are linked to the metabolites, M1_c and M1_m, and the flux distributions are shown in the figure. v_{fi} denotes as the forward fluxes, and v_{bi} are backward fluxes. The stoichiometric coefficients for the forward reaction are assumed to be 1, and -1 for the backward reactions.

Flux-sum synthesis rates for the metabolite, M1, in cytoplasm and mitochondrion are respectively obtained from the flux distribution.

$$r_i = \sum_{N_{ij}>0,j} N_{ij}v_{f,j} - \sum_{N_{ij}<0,j} N_{ij}v_{b,j}$$

$$r_{M1_c} = (1)v_{f1} + (1)v_{f2} = 1.2$$

$$r_{M1_m} = (1)v_{f3} - (-1)v_{b5} = 1.6$$

The overall flux-sum synthesis rates for the metabolite, M1:

$$r_m = \sum_{i \in \Omega^c} \left(\sum_{N_{ij} > 0, j} N_{ij} v_{f,j} - \sum_{N_{ij} < 0, j} N_{ij} v_{b,j} \right), m \in \Omega^m$$

$$r_{M1} = r_{M1_c} + r_{M1_m} = 2.8$$

Also, the mathematical formulation and the results do not transmit confidence in the proposed approach. For instance, Fig. 5 has a metabolite where the tiny ratio is likely to come from a solver error, or a problem with the tolerance, but the authors did not spot or correct this. Also, Figure 6 seems a clustering made on a single value from Figure 5, which is affected by the above problem.

The tiny ratio is not a solver error. We have provided a numerical example (supplementary material, S1) to illustrate definitions of logarithmic fold change ratio and similarity ratio. Logarithmic fold change ratio, $\log_2(r_{\text{deficient}}/r_{\text{normal}})$, is applied to present the difference between deficient and normal cases. As shown in the supplementary material, LFC for Metabolite 8 is zero because the flux-sum synthesis rate of deficient case is equal to that of the normal cell. LFCs for the template and 30 mutants are presented in Figure 6. The dark green or red colours indicate that flux-sum synthesis rate for the deficient is close to the normal cell.

On the outer and inner problems, the authors should clarify a major point: are the UFD problem and the FBA problem solved at the same level? If so, then this is actually a bilevel problem, and repeating the constraints $NV=0$ and $v_{LB} < v < v_{UB}$ is redundant. If not, then why are they not nested in the mathematical formulation? I can assume that the UFD is run as an outer problem compared to the FBA? Otherwise, how is the flux distribution calculated for the UFD? In any case, all these strategies (including norm-2 regularisation) have been adopted in the past in metabolic modelling, and the authors should clearly identify their contribution.

The FBA and UFD problems are hierarchically applied to compute flux distribution for each mutant. The maximum biomass growth rate, v^*_{biomaas} , obtained by the FBA is provided for a lower bound in the UFD problem to yield the uniform fluxes. Such fluxes are then supplied for evaluating the similarity ratio in the outer optimization problem.

The selection of the objective function is not clear. Are the authors optimising ATP and biomass with equal weight? Or is the objective selected according to the type of cancer/healthy model? If so, the authors should demonstrate that the flux distributions are comparable even when different objective functions are considered, and the results are not just driven by the change in the objective function. E.g., can the authors show or discuss that the effect of changing the objective function when

comparing flux or flux-sum distributions in healthy-vs-cancer is negligible compared to the actual difference of flux in healthy-vs-cancer?

We have revised the manuscript how to use the objective function in the FBA problem for computing the mutant and normal cases. The objective of the FBA is the maximization of the biomass growth rate for the cancer and mutant cells. However, normal cells may have different objectives depending on growth signals [24]; thus, the maximization of ATP synthesis rate v_{ATP} was applied in this situation.

The authors introduce similarity ratios. They should justify why this further discretisation of $u_m^{(T/W)}$ is needed, as it introduces another parameter (the tolerance). Is this discretisation the reason why the authors need to introduce the TLOP to solve the discretised problem? Could this be left as a continuous variable, avoiding the NHDE heuristic altogether? Why the NHDE is needed exactly, and how is it formulated (specifically, how are individuals defined?).

The similarity indicator is not for discretisation of the problem, but uses to figure out total numbers of similarity. We have provided a numerical example (supplementary material, S1) to illustrate definitions of similarity indicator. Indeed, TLOP is a mixed-integer optimization problem. The integer variables are applied to indicate which genes are selected to be mutated, and their corresponding regulation, e.g. up-regulation, down-regulation and knocked-out. NHDE can be straightforwardly applied to solve the problem.

Minor points

The manuscript has several typos, most of which could have been easily fixed with careful proofreading. E.g. “according literature survey”, “than do normal cells”

And infer that -> “and inferred that

Used the CORDA -> used CORDA

Figure 4 seems created with STRING, and the authors should mention it in the caption. Also, they refer to clusters as “first cluster”, “second cluster” etc., but the reader has no way to check the order of clusters from the figure.

We have corrected typos and revised the legend in Figure 7 (Original figure 4) to explain each cluster.

Figure 7. Protein–protein interactions of the inferred oncogenes. Protein–protein interactions (PPIs) of the inferred oncogenes with eight additional signalling proteins and their families, namely ACly, TP53, AKT, IGF, GSK, GLUT, PECK, and MTOR. *k*-means clustering function in STRING database was applied to classify the PPI network into four classes denoted by four colours. The proteins in the first cluster (green balls) participated in inositol phosphate and glycerophospholipid metabolisms, related to PTEN. The proteins in the second cluster (emerald green balls) were involved in triacylglycerol synthesis. The proteins in the third cluster (purple balls) were mainly involved in gluconeogenesis and amino sugar and citric acid cycle metabolisms. The proteins in the fourth cluster (red balls) were involved in other diversified metabolisms.

Reviewer: 2

Comments to the Author(s)

This is an interesting study to find metabolic oncogenes. The authors use a genome scale metabolic modeling approach to identify metabolic genes that when perturbed rewires normal cell metabolism towards cancer state. This is achieved by using a template cancer state and comparing the similarity of the metabolic state after gene knockdowns with the template. The approach is novel and useful for the cancer metabolism community.

Thank you for the positive reviews and constructive comments. You have spent a lot of time on this manuscript, and we appreciate their effort. We have carefully revised all recommendations, added the preliminary result for colon tissue for you to evaluate our approach.

Major revisions

One major concern is that the authors use one specific type of cancer (head and neck). But the driver genes they identify (e.g. PTEN) are not specific to that cancer. Many of the oncogenes are observed in other cancers and its unclear if they are unique to head and neck. I'd suggest that the authors repeat the analysis in 2 or three other cancer types to identify to those genes that are specific to a cancer type and those that are more general. Alternately, they should compare their predicted oncogenes with experimental data specific to head and neck cancer alone.

Our team is conducting wet-lab experiments using CRISPR and RNAi strategies to validate the identification. However, such an experiment should spend a lot of time. We will report the result in the future. We are applying the algorithm to reconstruct different cancer tissues, such as colon, lung and liver tissues, incorporated with HPA data to decipher their difference. The preliminary result for colon tissue is attached for you to evaluate our approach. We found the average similarity ratio is higher than 78%, but GNPDA1 and PGAM1 are infeasible. We will investigate the colon model in detail to include the information of PIP5K1.

Gene	Ave. SR
PTEN	0.788
PIKFYVE	0.839
PIK3CA	infeasible
PI4KA	0.966
TPK1	0.777

THTPA	0.841
G6PC	0.969
GNPDA1	infeasible
GFPT1	0.808
RFK	0.844
OPLAH	0.781
PGAM1	infeasible
PIP5K1	Does not included in the colon model

Most cancers have subtypes with distinct genomic features, including head and neck cancer. Can this data be stratified in to those sub types? Do those sub types have distinct metabolic activity? By lumping the sub types they may be mixing up signal from different sub types.

In this study, we applied the protein expressions of normal and cancerous head and neck tissues to reconstruct the tissue-specific genome-scale metabolic models. The protein expressions of the head and neck squamous cell retrieved from HPA are not discriminated into its subtypes so that the reconstructed models are not included the subtype information. The approach is accordingly incapable of inferring a subtype result.

The basal model doesnt make biological sense to me. The authors should have either used the entire recon2 model or started with the normal model and identified interventions that make it similar to cancer state. I can understand why they did it from a computational perspective but the biological significance of the basal model is unclear.

The BL model refers to as the normal model, which represents normal situations of head and neck squamous tissues. We have added this sentence in the revised manuscript to clear the definition of BL model. In this study, we acquired protein expressions in in human head and neck normal and tumour tissues, respectively, from HPA. Such data were then provided to reconstruct HT model and CA model, respectively. The HT model consisted of 2256 metabolites and 3427 reactions, and CA model comprised 2147 metabolites and 3253 reactions. The two models were combined to form a basal/normal model for investigating how a healthy cell can smoothly detour metabolic reprogramming to that of a cancer cell.

The authors haven't rigorously pursued the clinical significance of the metabolic alterations. Does downregulation or mutation in these enzymes lead to reduced survival? If these genes are not associated with patient survival, then they may not be oncogenic drivers.

We have not rigorously clinical trials to investigate survival analysis for the inferred oncogenes. The survival investigation needs several years to achieve the result. In the revision, we surveyed survival analysis from HPA to explain survival significance of inferred oncogenes in Table 1, and the detailed results were also shown in the supplementary file (Table S3).

There's no statistical validation of the predictions. There are numerous metabolic oncogenes. So it is unclear if the genes identified by the authors could have been arrived at by random chance. They should do a sampling analysis (of metabolic genes in the model) and identify overlap with all known metabolic oncogenes. The overlap should then be compared with the overlap obtained by their approach.

We have added flux variability analysis in the revised manuscript to clarify such a bias result. The FBA in the TLOP problem was employed to evaluate flux alterations for normal and cancer model. However, it is a bias method to compute the flux pattern. Flux variability analysis (FVA) can be applied in a posterior inspection to overcome such a drawback, but required a lot of computation time. We applied the FVA to compute the minimum and maximum quantities of each metabolite for the normal model and the mutants, respectively. The flux intervals of each deficient case were compared with the normal intervals, and classified into seven categories by using the definition presented in the electronic supplementary material S1. We evaluated the trends of flux and flux-sum synthesis rate for each metabolite of the mutants and those of the template, and then counted the number of metabolites with same flux/flux-sum trends in the flux interval for each mutant and the template to obtain similarity ratios as shown in Figure 5. We found that the flux-sum similarity ratios and flux similarity ratios of all mutants are greater than 47% and 65%, respectively, so that the average ratio for each mutant is higher than 56%.

Fig 5. Similarity ratios of flux and flux-sum synthesis rates evaluated by flux variability analysis in a posterior inspection. The similarity ratio for each mutant is the ratio of the number of metabolites with same flux/flux-sum trends in the flux interval of the mutant to those of the template.

Reviewer: 3

Comments to the Author(s)

In this work, the authors utilized Recon2.2 (a human metabolic network reconstruction) and the Human Protein Atlas database in order to reconstruct a head and neck squamous cell-specific network describing both the healthy and the cancer cells. They used the Cost Optimization Reaction Dependency Assessment (CORDA) algorithm proposed by Schultz and Qutub to build the genome-scale models in a tissue-specific manner. On top of that, they developed a tri-level optimization framework to identify dysregulated enzymes in this cancer type. This is an interesting work that deserves to be published. However, there are a number of minor issues that would need to be addressed before the manuscript can be suitable for publication.

Thank you for the positive reviews and constructive comments. You have spent a lot of time on this manuscript, and we appreciate their effort. We have carefully revised all recommendations, added the abbreviation section in the revised manuscript to clarify the meaning of each acronym and provided a supplementary material to explain the algorithm.

Overall, there are too many acronyms in the manuscript, which make it hard to follow. Furthermore, as might not all readers be familiar with the terminology used in metabolic models it would be nice if some definitions are given i.e. choke-point metabolites, flux-sum analysis, as well as other terms used in this manuscript like templates, mutant flux patterns, etc.

We have added the abbreviation section in the revised manuscript to clarify the meaning of each acronym. We have provided the supplementary material, S1 to illustrate definitions of flux-sum, logarithmic fold change ratio, similarity ratio and choke-point metabolites.

It would be nice if, at the end of the introduction where the current work is described, you could add more the intuition between the algorithms used instead of repeating the algorithmic steps. Furthermore, the tri-level optimization algorithm in Methods should be better explained. Maybe you could give the overall idea by describing each optimization problem and then become more mathematically specific. I was always come back and forth in order to understand it.

We have revised the manuscript and redrawn Figure 2 (as shown in below) to explain the design concept of the algorithm. We have additionally explained the NHDE algorithm in the supplementary material, S1.

TLOP is applied to mimic mutant schemes in a wet lab for oncogene inference. The outer optimization problem was employed to decide mutation genes and their

corresponding dysregulated bounds. The decision variables were used in the FBA and UFD problems to hierarchically compute flux distribution for each mutant. The fluxes obtained from the decision variables in FBA and UFD problems were then applied to evaluate the similarity ratios in the outer problem. The evolutionary computation was repeatedly until the optimal result achieved.

More specifically,

Introduction:

-Please rephrase: “Flux-dependent and pruning methods are automated tissue-specific reconstruction algorithms for generating a genome-scale metabolic model for predicting tissue-specific behaviours [6, 10]. “

We have rephrased the sentence as “Flux-dependent and pruning methods are applied to reconstruct tissue-specific genome-scale metabolic models for predicting tissue-specific behaviours”.

- A tri-level inference optimization framework: Say here what it does in a few words... Please, give the underlying intuition.

We have rewritten the sentence in the revised manuscript as follows: A tri-level inference optimization framework integrated the templates and CBM was developed to infer dysregulated enzymes that contribute to inducing head and neck squamous cell carcinoma (HNSCC). Such framework can also be used to mimic gene screening procedures in wet lab for detecting dysregulated oncogenes.

- HNSCC: I think this is redundant considering that you have already defined HNSCs.

Abbreviations of HNSCC and HNSC are different. To avoid such a confusion, we keep HNSCC only.

HNSCC: Head and neck squamous cell carcinoma

HNSC: Head and neck squamous cell

Methods:

- HNSC: use the acronym instead.

- definition or reference of flux-sum alterations

- “The basal model (union set of CA and HT models) was used”: better move the definition at the beginning where you first mention it.

- “TLOP was used to simulate a wet lab...” what exactly do you mean by that?
- chokepoint metabolites: give the definition -“The objective function in the outer optimization problem is a similarity measure that...” Please make clearer what the outer objective does. Give the intuition behind.
- “the Warburg hypothesis obtained from the literature...” What exactly do you obtain from the literature? Please, be more precise. Please, add the related references.
- Shouldn't the definition of SR go first together with its meaning in order to understand the trilevel optimization?
- the logarithmic fold changes: Add the acronym LFC where you first mention it. Not later.
- UFD is defined later on page 13.

We have added the abbreviation section in the revised manuscript to clarify the meaning of each acronym, and provided the supplementary material, S1 to illustrate definitions of flux-sum, logarithmic fold change ratio, similarity ratio and choke-point metabolites.

Results and Discussion

-section 3.1: shouldn't this section be presented earlier in the manuscript in methods... it helps a lot the overall understanding.

In section 2.1, we described the reconstruction method that is not only to build a genome-scale metabolic model of head and neck tissue, but also apply to other tissues. In section 3.1, we presented the reconstructed results for head and neck tissue.